# Effect of Supplementation of Cryoprotectant Solution with Hydroxypropyl Cellulose for Vitrification of Bovine Oocytes

**DOI:** 10.3390/ani12192636

**Published:** 2022-09-30

**Authors:** Min-Jee Park, Seung-Eun Lee, Jae-Wook Yoon, Hyo-Jin Park, So-Hee Kim, Seung-Hwan Oh, Do-Geon Lee, Da-Bin Pyeon, Eun-Young Kim, Se-Pill Park

**Affiliations:** 1Faculty of Biotechnology, College of Applied Life Sciences, Jeju National University, 102 Jejudaehak-ro, Jeju-si 63243, Korea; 2Stem Cell Research Center, Jeju National University, 102 Jejudaehak-ro, Jeju-si 63243, Korea; 3Mirae Cell Bio, 1502 ISBIZ Tower, 147 Seongsui-ro, Seongdong-gu, Seoul 04795, Korea

**Keywords:** bovine oocytes, cryoprotectant, hydroxypropyl cellulose, solution vitrification

## Abstract

**Simple Summary:**

This study investigated the effects of the addition of hydroxypropyl cellulose (HPC) to a vitrification solution of bovine oocytes. For the vitrification, bovine metaphase II oocytes were pretreated with a solution containing 10% ethylene glycol supplemented with 0, 10, 50, or 100 µg/mL HPC for 5 min, then exposed to a solution containing 30% ethylene glycol supplemented with 0, 10, 50, or 100 µg/mL HPC for 30 sec, and then directly plunged into liquid nitrogen. The survival rate of the oocytes was significantly higher in the 50 HPC group than in the 0, 10, and 100 HPC groups. The reactive oxygen species (ROS) levels of the non-vitrified-thawed (VT) and the 50 HPC groups were significantly lower than the 0, 10, and 100 HPC groups. The mRNA expression levels of the genes related to proapoptotic (Bax) was higher in the 0 and 100 HPC groups than in the 50 and non-VT groups. The mRNA levels of antiapoptotic genes (BCl2) were higher in the non-VT than in the other groups. The development rates of the embryos (day 8) obtained via parthenogenetic activation (PA) were determined in the non-VT, 0 HPC, and 50 HPC groups. The development rate was significantly higher in the non-VT and 50 HPC groups than in the 0 HPC group. Conclusion: Supplementation of vitrification solution with HPC improves the survival of VT bovine oocytes and the development capacity of embryos derived from these oocytes via PA.

**Abstract:**

The technology of successful cryopreservation is a very important factor in research and commercial applications. However, the survival and development of the vitrified-thawed (VT) oocytes are lower than those of non-vitrified-thawed (non-VT) oocytes. This study investigated the effect of the addition of hydroxypropyl cellulose (HPC) to a vitrification solution of bovine oocytes. For the vitrification, bovine metaphase II oocytes were pretreated with a solution containing 10% ethylene glycol supplemented with 0, 10, 50, or 100 µg/mL HPC for 5 min, then exposed to a solution containing 30% ethylene glycol supplemented with 0, 10, 50, or 100 µg/mL HPC for 30 sec, and then directly plunged into liquid nitrogen. Oocytes exposed to 0, 10, 50, and 100 µg/mL HPC were named the 0, 10, 50, and 100 HPC groups, respectively. Samples were thawed via sequential incubation in Dulbecco’s phosphate-buffered saline (D-BPS) supplemented with 10% fetal bovine serum and decreasing concentrations of sucrose (1, 0.5, 0.25, and 0.125 M) for 1 min each time. After thawing, VT oocytes were treated at 0.05% hyaluronidase, and cumulus cells were removed by mechanical pipetting. The oocytes were washed with HEPES-buffered Tyrode’s medium and incubated in a droplet of previously cultured in vitro maturation medium for 1 h to recover. The survival rate of the oocytes was significantly higher in the 50 HPC group (84.2%) than in the 0 (75.4%), 10 (80.4%), and 100 (75.5%) HPC groups. The reactive oxygen species (ROS) levels of the non-VT and 50 HPC groups were lower than the 0, 10, and 100 HPC groups. The mRNA levels of proapoptotic genes (*Bax*) were lower in the non-VT, 0, and 50 HPC groups than in the other groups. The mRNA expression levels of antiapoptotic genes (*BCl2*) was higher in the non-VT than in the other groups. The mRNA level of a stress-related gene (*Hsp70*) was lower in the 50 HPC than in the other groups. At day 8, the developmental capacity of embryos obtained via parthenogenetic activation (PA) was determined in the non-VT, 0 HPC, and 50 HPC groups. The cleavage rate of the non-VT group was significantly higher, but the blastocyst development rate and total cell number per blastocyst did not significantly differ between the non-VT and 50 HPC groups. The mRNA levels of proapoptotic genes (*Bax* and *Caspase-3*) and a stress-related gene (*Hsp70*) were higher in the 0 HPC group than in the non-VT and 50 HPC groups. In conclusion, supplementation of vitrification solution with HPC improves the survival rate of VT bovine oocytes and the development capacity of embryos derived from these oocytes via PA.

## 1. Introduction

Cryopreservation ensures a consistent and steady supply of oocytes. Oocyte cryopreservation is useful not only for long-term storage of female genetic materials, but also for reproduction of endangered species and livestock with high economic value. It is particularly useful for the production of embryos via in vitro fertilization (IVF) and somatic Checked itcell nuclear transfer (SCNT). Oocyte quality greatly affects embryo development in vitro. Production of a parthenogenetic embryo is more efficient than SCNT embryo production efficiency, since it goes through a process of PA when producing SCNT embryos; if a positive effect on the development rate of PA embryos is confirmed using HPC-treated VT eggs, it is judged that the results can be applied to improve the SCNT embryo development rate.

Oocyte vitrification has been successfully used to cryopreserve oocytes of humans and many other species. Various technologies have been described for the vitrification of bovine [1], pig [2], mouse [3], and human [4] oocytes. However, oocytes are inevitably damaged during vitrification.

Previous studies regarding the effect of cryopreservation on in vitro matured oocytes reported that vitrified-thawed (VT) oocytes have a normal spindle and karyotype, but exhibit cortical granule exocytosis, swelling of smooth endoplasmic reticulum vesicles, and mitochondrial damage [5,6]. In general, oocytes are more prone to damage during the cooling process than embryos because the integrity of the metaphase spindle microtubules is perturbed during cooling and high concentrations of cryoprotective agents damage oocytes [7]. In addition, the mechanism underlying the vitrification of oocytes is complex and likely involves disruption of cellular structures leading to the premature release of cortical granules, damage to zona hardening, damage of normal fertilization, and changes in apoptosis and oocyte gene expression [8]. Oocyte damage upon freezing negatively affects the development of embryos generated via IVF and SCNT. Therefore, this study attempted to minimize oocyte damage during vitrification by supplementing the vitrification solution with the synthetic polymer hydroxypropyl cellulose (HPC) and investigated the effects on the development of embryos generated via parthenogenetic activation (PA).

HPC is a polysaccharide with a variable length that can form a viscous gel at low temperatures because it has very similar physical properties as albumin-based formulations with certain molecular weight [9]. It is listed in a pharmacopeia and can be safely used because it is a general food additive and drug excipient [10]. Recent studies reported that HPC increases the viscosity of the vitrification solution and reduces the amount of time that an embryo spends attached to the Cryotop surface during thawing, indicating that it can improve vitrification by reducing the risk of cryogenic injury and increasing the survival rate [10]. HPC has been added to the vitrification solution for freezing human oocytes [11] and embryos [10]. This study investigated the effect of supplementation of vitrification solution with HPC on the vitrification of bovine oocytes and the developmental capabilities of embryos cultured from these oocytes via PA. The objective of this study was to modify this method using HPC as a supplement to improve vitrification.

## 2. Materials and Methods

### 2.1. Experiment Design

To determine the appropriate concentration of HPC, the oocytes were exposed to 0, 10, 50, and 100 µg/mL HPC (called the 0, 10, 50, and 100 HPC groups, respectively). The concentration of HPC was determined by referring to a previously published paper [11]. HPC (pharmaceutical standard, 80,000 Da average molecular weight, Sigma Aldrich) was diluted with H_2_O at room temperature. We investigated the survival rate, reactive oxygen species (ROS) level, and mRNA expression (*Bax*, *Bcl2*, *Hsp70*, and *Dnmt3a*) of non-VT oocytes and oocytes in the 0, 10, 50, and 100 HPC groups; each group used about 200 oocytes. The survival rate of the 50 HPC group was the highest, and thus PA embryo development was investigated for the non-VT, the 0 HPC, and 50 HPC groups; in the experiment, 147–170 oocytes were used for each group. Next, we parthenogenetically activated the oocytes and determined the development rate at days 2 and 8, the total cell number per blastocyst, and the number of apoptotic cells. We compared the mRNA expression levels of genes (*Glut-5, Interferon-tau, Caspase-3, Hsp70, HSF, Bax,* and *Dnmt3a*) related to the developmental potential between blastocysts produced from non-VT oocytes and those produced from oocytes in the 0 and 50 HPC groups.

### 2.2. Oocyte Preparation and In Vitro Maturation (IVM)

Bovine ovaries were transported from a slaughterhouse to the laboratory in a saline buffer (9 g/L NaCl and 0.1 g/L penicillin–streptomycin). The temperature of the thermos used for transportation was maintained at 30–33 °C. Cumulus-oocyte complexes (COCs) were aspirated from visible follicles measuring 2–6 mm using an 18-gauge needle and washed with an HEPES-buffered Tyrode’s medium (TL-HEPES). They were cultured in an incubator in IVM medium, which comprised TCM-199 (Gibco, Grand Island, NY, USA) containing 10% fetal bovine serum (FBS, Grand Island, NY, USA), 0.2 mM sodium pyruvate, 1 µg/mL follicle-stimulating hormone (Folltropin™; Bioniche Animal Health, Belleville, ON, Canada), 1 µg/mL estradiol-17β, and 25 µg/mL gentamycin sulfate. Sets of 10 COCs were matured in 50 µL IVM medium under mineral oil at 38.8 °C in a humidified atmosphere of 5% CO_2_ in air for 20–22 h.

In this experiment, 30–100 bovine oocytes were matured 30 times, and a total of about 2000 oocytes were matured.

### 2.3. Vitrification and Thawing

A D-PBS (Sigma) solution with 10% FBS was used for the pretreatment, vitrification, and dilution. The pretreatment solution included 10% ethylene glycol (EG). The vitrification solution included 30% EG and 0.5 M sucrose. 

Similarly to the vitrification solution, the thawing solution used D-PBS with 10% FBS and sucrose of each concentration (1, 0.5, 0.25, or 0.125 M). VT oocytes were diluted sequentially from high to low concentrations. About 1500 oocytes were vitrified-thawed according to the previously reported MVC method [12]. After 20 to 22 h of IVM, metaphase II (MII) oocytes were treated in 0.1% hyaluronidase and mechanically pipetted to remove cumulus cells. Oocytes were washed 3 times with TL-HEPES and incubated in a pre-cultured IVM solution droplet for 1 h to recover. Freezing procedures were performed at room temperature. After washing 3 times in TL-HEPES, MII oocytes were treated with D-PBS for 5 min. For the vitrification, 1500 oocytes were pretreated with EG10 containing 0, 10, 50, or 100 µg/mL HPC for 5 min, exposed to EG30 containing 0, 10, 50, or 100 µg/mL HPC for 30 sec, and then the VT oocytes were loaded individually toward the wall of the French Ministraw, which was 2.5–3.0 cm that had been covered with a minimum volume of vitrification solution. After plunging the straws in liquid nitrogen, 4–5 straws were put into a pre-chilled cryovial, stored in a cane, and placed in a liquid nitrogen container. Cryopreservation solution was removed via a five-step process using a solution heated to 37 °C for thawing. Straws stored in the liquid nitrogen container were quickly transferred to a solution containing 1.0 M sucrose in D-PBS and treated at room temperature for 1 min 30 sec. Oocytes were sequentially transferred to D-PBS containing sucrose of 0.5, 0.25, and 0.125 M, treated for 1 min, and then transferred to D-PBS without sucrose. Finally, the VT oocytes were incubated in TCM-199 with 10% FBS for 1 h.

### 2.4. PA and In Vitro Culture

Non-VT and VT oocytes were activated with 10 µM calcium ionophore in Charles Rosenkrans 1 medium with amino acids (CR1aa) containing 3 mg/mL bovine serum albumin (BSA) for 5 min, immediately placed in 2 mM 6-dimethylaminopurine, and incubated under mineral oil at 38.8 °C in a humidified atmosphere of 5% CO_2_ in air for 3 h. After activation, 20–40 embryos were cultured per 50 µL droplet of CR1aa medium with 0.03% BSA (no fatty acid) for 96 h. On day 4, 10–20 embryos were switched to a 10 µL droplet of CR1aa medium containing 10% FBS and incubated for a further 4 days in an incubator containing 5% CO_2_. at 38.8 °C. Since the 50 HPC group showed the highest survival rate among the HPC treatment groups, embryo development was examined in the non-VT, 0 HPC, and 50 HPC groups.

### 2.5. Measurement of Intracellular ROS

Intracellular ROS was measured in oocytes by the 2,7-dichlorofluorescein assay used in the previous study [13]. Oocytes were cultured at 38.8 °C for 20 min with 100 µM 2,7-dichlorodihydrofluorescein diacetate (DCHFDA), washed 3 times in TL-HEPES to remove excess dyes, and analyzed immediately using an epifluorescence microscopy at 450–490 nm and 515–565 nm wavelength. A digital camera was attached to a microscope to acquire grayscale images. ImageJ (NIH, Bethesda, MD, USA) was used to measure the mean values. Prior to the statistical analysis, the background fluorescence value was subtracted from the value. In total, 10 to 20 oocytes were experimented with at one time, and the number of repetitions was three times.

### 2.6. TUNEL Assay

An investigation was performed using the In Situ Cell Death Detection Kit (Roche, Mannheim, Germany) on the number of apoptosis cells in the blastocyst produced by PA on Day 8. Briefly, 10–15 blastocysts in each group were fixed in phosphate-buffered saline containing 3.7% paraformaldehyde for 1 h and then the blastocysts were permeabilized in 0.3% Triton X-100 for 1 h and incubated with fluoresce-in-conjugated deoxyuridine triphosphates and terminal deoxynucleotidyl transferase in a place with no light for 1 h. This process was incubated in 50 µg/mL RNase A for 1 h at 37 °C. The nuclei were simultaneously counterstained with 40 µg/mL propidium iodide. The stained blastocyst loaded on the glass slide was observed in a fluorescence microscope equipped with a UV filter. Respectively, red indicates chromatin, green indicates fragmented DNA combined, and yellow (merged red and green) indicates fragmented DNA of an apoptotic blastomere. The apoptotic index was decided as the percentage of yellow blastomeres among the total number of red blastomeres.

### 2.7. Real-Time RT-PCR 7

Real-time RT-PCR was performed as described previously [13]. The primers are described and listed in Table 1. A Dynabeads mRNA Direct Kit (Invitrogen, Carlsbad, CA, USA) was used to prepare the mRNA from 20 oocytes and 20 blastocysts per group. cDNA was synthesized using oligo(dT)20 primer and SuperScript III reverse transcription enzyme (Invitrogen) and from 2 μg mRNA per sample. Real-time RT-PCR was performed on a StepOnePlus real-time PCR system (Applied Biosystems, Warrington, UK) and the final reaction volume was 20 µL, which contained SYBR Green PCR Master Mix (Applied Bio-systems). The PCR conditions were as follows: 10 min at 95 °C, followed by 40 cycles of 15 sec at 95 °C and 60 sec at 50–60 °C. Samples were then cooled to 12 °C. Relative gene expression was analyzed by the 2^−ΔΔCt^ method [14]. Expression levels were normalized against the mRNA level of *β-actin*. The experiment was independently repeated 4–5 times.

### 2.8. Statistical Analysis

All data analysis used the general linear model procedure included in the Statistical Analysis System (SAS User’s Guide, 1985, Statistical Analysis System Inc., Cary, NC, USA). Significant differences were determined by Tukey’s multiple range test. The comparison of relative gene expression used the paired Student’s *t*-test. *p*  <  0.05 was regarded as statistically significant.

## 3. Results

### 3.1. Comparison of the Survival Rate and ROS Level of Oocytes between the Various Groups

The survival rates of VT bovine MII oocytes treated with vitrified solutions containing different concentrations of HPC are shown in Table 2. The survival rate was 75.4%, 80.4%, 84.2%, and 75.5% in the 0, 10, 50, and 100 HPC groups, respectively, and was significantly higher in the 50 HPC group than in the other groups. The morphologies of oocytes in the 0, 10, 50, and 100 HPC groups are shown in Figure 1. There was no marked morphological difference between the groups under a microscope. The level of ROS was compared between the non-VT and VT groups (Figure 2A). There was no significant difference in the ROS levels of the non-VT and 50 HPC groups, and the 0, 10, and 100 HPC groups were significantly lower. (Figure 2B).

### 3.2. Comparison of mRNA Expression Levels of Oocytes between the Various Groups

mRNA expression levels of genes related to apoptosis (Bax), antiapoptotic (Bcl2), stress (Hsp70), and demethylation (Dnmt3a) were compared between VT and non-VT oocytes. The mRNA expression level of Bax was significantly higher in the 0 and 100 HPC groups than in the non-VT, 10 HPC, and 50 HPC groups. The Bcl2 gene expressed mRNA level was significantly higher in the non-VT than in the 0 and 10, 50, and 100 HPC groups. The mRNA expression level of Hsp70 was significantly lower in the 50 HPC group than in the non-VT and 0, 10, and 100 HPC groups. However, the mRNA expression level of Dnmt3a did not differ between the groups (Figure 3).

### 3.3. Effect of HPC Treatment during Freezing of Oocytes on Development of Embryos Obtained via PA

The developmental potentials of embryos derived from non-VT and VT oocytes following PA were compared. The percentage of thawed oocytes that survived was higher in the 50 HPC group (80.1%) than in the 0 HPC group (74.0%) (Table 3). At day 2 post-activation, the cleavage rate of the embryos was significantly higher in the non-VT group (94.6%) than in the 0 (74.7%) and 50 (75.2%) HPC groups but did not significantly differ between the 0 and 50 HPC groups. At day 8 post-activation, the percentage of embryos successfully developed up to the blastocyst was higher in the non-VT group (21.1%) than in the 0 (7.1%) and 50 (13.0%) HPC groups and was significantly higher in the 50 HPC group than in the 0 HPC group. The mean total cell number of blastocysts was significantly higher in the non-VT (120.2 ± 6.4) and 50 HPC (100.2 ± 10.0) groups than in the 0 HPC group (82.7 ± 2.5). The apoptotic index of blastocysts was determined by fluorescence microscopy and the TUNEL assay. It was lower in the non-VT group (1.7%) than in the 0 (4.8%) and 50 (4.0%) HPC groups. There was no significant difference in the apoptotic index of the 0 and 50 HPC groups (Table 3). The morphologies of the embryos at day 2 and the blastocysts at day 8 in the non-VT, 0 HPC, and 50 HPC groups are shown in Figure 4.

### 3.4. Comparison of the mRNA Expression Levels of Genes Related to Developmental Potential in Blastocysts between the Groups

The mRNA expression levels of genes (*Glut-5, Interferon-tau, Caspase-3, Hsp70, HSF, Bax,* and *Dnmt3a*) related to the developmental potential in in vitro produced blastocysts generated via PA were compared between the groups. The mRNA expression levels of Dnmt3a (methylation-related) and HSF (stress-related) did not differ between the non-VT, 0 HPC, and 50 HPC groups. The mRNA expression levels of the proapoptotic genes Bax and Caspase-3 were significantly higher in the 0 HPC group than in the non-VT and 50 HPC groups. The mRNA expression levels of Glut-5 (metabolism-related) and Interferon-tau (implantation-related) of the 0 and 50 HPC groups were significantly lower than non-VT, but the mRNA level of Glut-5 in the 50 HPC group was higher than the 0 HPC group. The mRNA level of Hsp70 (stress-related) was remarkably lower in the non-VT and 50 HPC groups than in the 0 HPC group (Figure 5).

## 4. Discussion

Cryopreservation enables a stable supply of oocytes for the production of embryos via IVF and SCNT. IVF is a basic technology used for animal production, while SCNT is a useful technology for the preservation of endangered species and the production of transgenic animals. In addition, cryopreservation of human oocytes has advantages over cryopreservation of human embryos. Specifically, it preserves the fertility of women who are at risk of becoming infertile due to diseases, facilitates the donation of oocytes, and allows women to choose when they have children. Furthermore, the freezing of oocytes eliminates the legal and ethical concerns associated with the freezing of embryos. During cryogenic retention, cells are exposed to numerous mechanical, thermal, and chemical stressors [15], which can lead to cell dysfunction and cell death. In general, oocytes are more sensitive to damage caused by exposure to low temperatures than embryos [16]. Oocytes exhibit low permeability to water and cryopreservation solution due to their large size and are, therefore, very sensitive to low temperature-mediated preservation [17]. To solve these problems, vitrification is a simple, fast, cost-effective, and dependable method, and many vitrification methods have been reported [18,19]. We previously studied the suitability of the MVC method for vitrification [12] and developed an efficient vitrification method for the production of SCNT embryos [13]. Based on these studies, we sought to identify a supplement that can be added to the vitrification solution and improve the survival rate of bovine oocytes and the development ability of embryos derived from these oocytes. HPC increases the viscosity of vitrification solutions and reduces the risk of cryodamage by improving the efficiency of solidification during the vitrification process [10]. We added HPC to vitrification solution and investigated the survival rate of VT oocytes and embryo development after PA.

We supplemented the vitrification solution with 0, 10, 50, and 100 µg/mL HPC in an attempt to increase the survival rate of bovine oocytes. The survival rate of VT oocytes was significantly higher than that in the 0, 10 and 100 HPC groups. ROS are byproducts of normal mitochondrial metabolism and major signaling molecules in various physiological and pathological processes and mediated oxidative stress [20]. An increase in the level of ROS can reduce the intracellular adenosine triphosphate concentration and the glutathione/glutathione disulfide ratio, and concomitantly increase the cytosolic concentration of calcium ions, which can damage oocytes. We expected that freezing would affect the ROS level of the oocytes. The ROS level was lowest in the 50 HPC group among the HPC treatment groups and was similar to the non-VT group. This result suggests that the levels of exposure to oxidative stress appear differently in various groups of oocytes, and that among the VT oocytes, those in the 50 HPC group were exposed to the lowest level of oxidative stress. To investigate damage of the VT oocytes and the effect of HPC, the mRNA expression levels of Bax and Bcl2 (apoptotic-related), Hsp70 (stress), and Dnmt3a (methylation) were determined. Bax regulates mitochondrial membrane permeability and induces apoptosis by disrupting the mitochondrial membrane [21], while Bcl2 is an anti-apoptotic protein that promotes survival [22]. Expression of Bax is increased [23] and expression of Bcl2 is decreased [24] after vitrification. Alterations in the expression of apoptosis-related genes in oocytes upon vitrification may become involved with embryo development because the majority of apparently normal oocytes unsuccessfully develop during the first few days of culturing after vitrification and degenerate [25]. In this study, mRNA expression of Bax did not significantly differ between the non-VT, 10 HPC, and 50 HPC groups, but was higher in the 0 and 100 HPC groups than in the non-VT group. mRNA expression of Bcl2 did not significantly differ between the non-VT and HPC-treated groups but was significantly lower in the 0 HPC group. These results suggest that HPC prevents apoptosis. The mRNA expression of the Hsp70 gene (stress-related) in the 50 HPC group was significantly lower than that of the non-VT group. We expected Hsp70 expression to increase upon vitrification due to oocyte damage, and further research is required to elucidate the mechanism by which HPC reduces Hsp70 expression. Oocyte vitrification is accompanied by alterations in DNA methylation in oocytes and embryos, which may make embryo development difficult and explain the reduced embryo development quality observed [26,27]. A previous study reported that Dnmt3a expression is significantly reduced in vitrified oocytes [28]. However, we found that Dnmt3a expression did not significantly differ between non-VT and VT oocytes.

To investigate the effect of HPC on embryo development, oocytes in the non-VT, 0 HPC, and 50 HPC groups underwent PA and were cultured for 8 days. The cleavage rate at day 2, blastocyst formation rate at day 8, and total cell number per blastocyst significantly differed between the non-VT and VT groups. The cleavage and blastocyst formation rates were significantly higher in the non-VT group. The cleavage rate did not significantly differ between the 0 and 50 HPC groups, but the blastocyst formation rate was higher in the 50 HPC group (13.0%) than in the 0 HPC group (7.1%). The cleavage rate of reconstructed embryos generated using VT oocytes was reported to be ~50–70% [29,30], while the blastocyst formation rate was reported to be ~4–8% [31,32,33]. In these previous studies, it was reported that the development rate of the blastocyst stage obtained by reconstructed embryos via PA was 7–13%. The TUNEL assay found no significant difference in apoptosis between the 0 and 50 HPC groups, but proapoptotic gene expression levels were significantly higher in the 0 HPC group than in the non-VT and 50 HPC groups. VT oocytes are repeatedly exposed to heat shock stress. The heat shock gene induces apoptosis in preimplantation, which uses a developmentally regulated manner [34]. mRNA expression of the Hsp70 gene was significantly lower in the non-VT and 50 HPC groups than in the 0 HPC group. This is similar to the differences in mRNA expression of Caspase-3 and Bax (proapoptotic). These results indicate that the supplementation of vitrification solution of oocytes with HPC reduces stress in embryos derived from these oocytes via PA. To investigate the developmental potential of embryos obtained via PA, the expression level of various factors was determined. The mRNA expression level of the Glut-5 gene (metabolism-related) was highest in the non-VT group, but the 50 HPC group was higher than the 0 HPC group. mRNA expression of Dnmt3a did not significantly differ between non-VT and VT oocytes. These results indicate that oocyte damage caused by vitrification perturbs embryo development. Similar results have been previously reported [13]. However, the mRNA expression level between each group was similar in this study. This indicates that HPC does not markedly affect DNA methylation, but further research is needed. Supplements must be identified that reduce oocyte damage during vitrification and thawing, and thereby improve embryo development in vitro. We propose that HPC is one such supplement.

In conclusion, this study demonstrates that supplementation of vitrification solution with HPC improves the survival rate of VT oocytes, decreases ROS production, reduces cell death, and improves the development of embryos obtained via PA. Our results indicate that HPC is suitable for use as a vitrification solution supplement.

## Figures and Tables

**Figure 1 animals-12-02636-f001:**
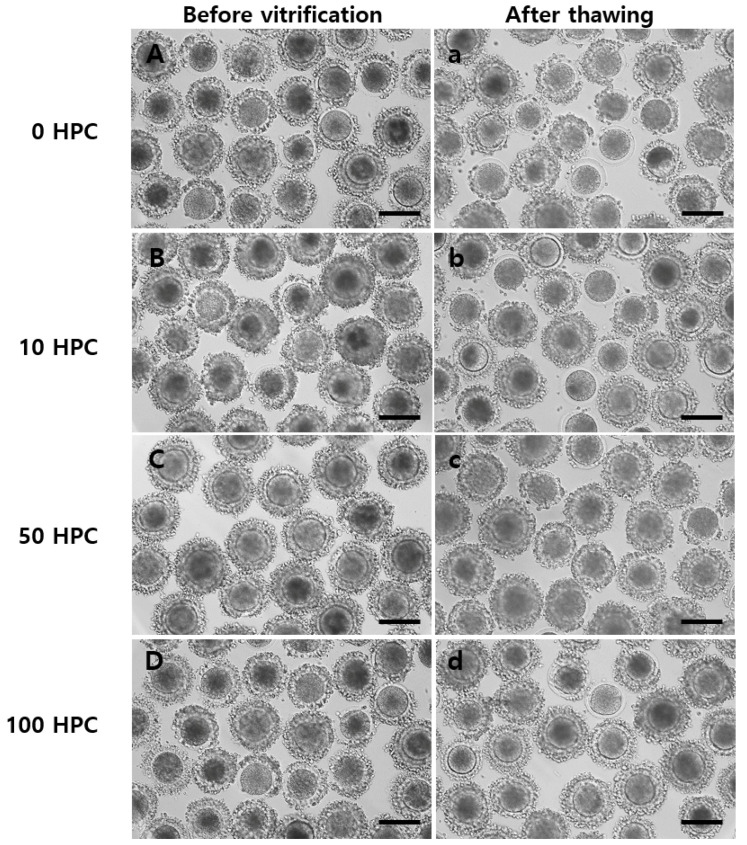
Morphologies of oocytes in the 0 (**A**,**a**), 10 (**B**,**b**), 50 (**C**,**c**), and 100 (**D**,**d**) HPC groups. (**A**–**D**): before vitrification, (**a**–**d**): after thawing. Bar, 100× in (**A**–**D**) and (**a**–**d**).

**Figure 2 animals-12-02636-f002:**
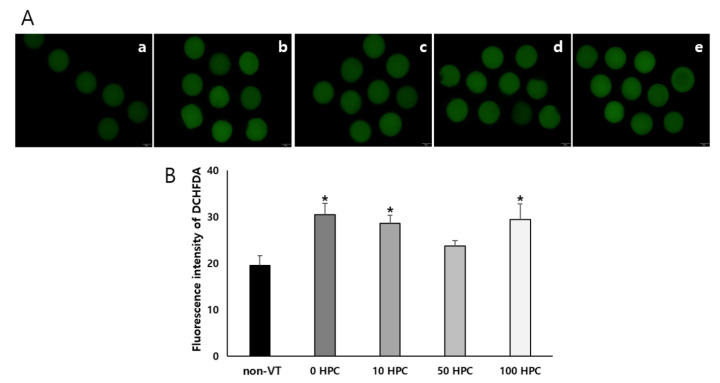
Comparison of levels of ROS in HPC (0, 10, 50, and 100) groups and the non-VT group. Fluorescence images of ROS (**A**,**a**–**e**). ROS activity was detected with DCHFDA (green). Fluorescence intensity of DCHFDA (**B**). * *p* < 0.05 compared with the non-VT group.

**Figure 3 animals-12-02636-f003:**
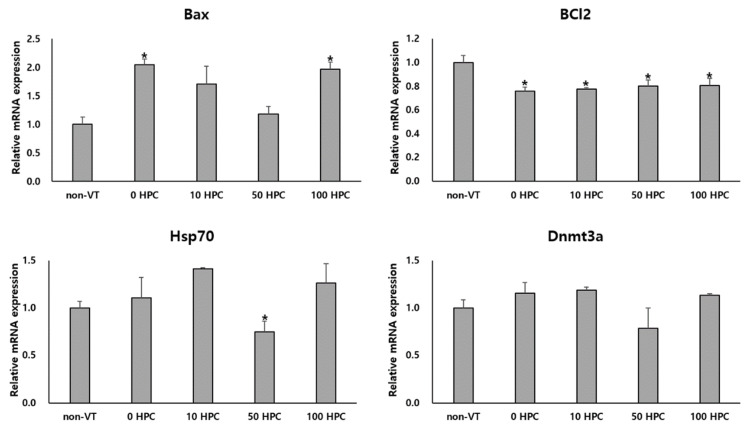
Relative mRNA expression levels of genes related to apoptosis (Bax and Bcl2), demethylation (Dnmt3a), and stress (Hsp70) in non-VT and VT oocytes. * *p* < 0.05 compared with the non-VT group.

**Figure 4 animals-12-02636-f004:**
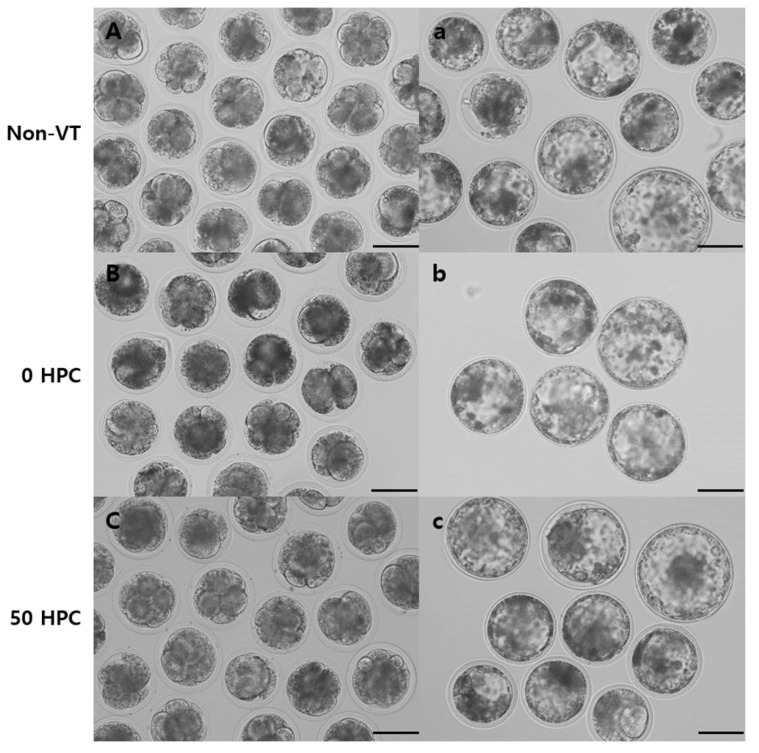
Morphologies of embryos obtained via PA in the non-VT, 0 HPC, and 50 HPC groups. Images of cleaved embryos at day 2 ((**A**), non-VT; (**B**), 0 HPC; and (**C**), 50 HPC) and blastocysts at day 8 ((**a**), non-VT; (**b**), 0 HPC; and (**c**), 50 HPC). Bar, 100 µm.

**Figure 5 animals-12-02636-f005:**
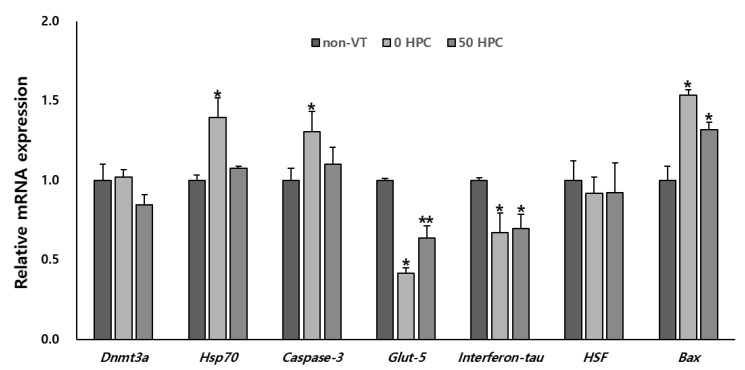
mRNA expression levels of genes (*Dnmt3a*, *Hsp70*, *Caspase-3*, *Glut-5*, *Interferon-tau*, *HSF*, and *Bax*) related to development potential in blastocysts produced via PA in the non-VT, 0 HPC, and 50 HPC groups. The internal standard used β-actin gene. * *p* < 0.05 compared with the non-VT group. ** *p* < 0.05 compared with the non-VT and 0 HPC groups.

**Table 1 animals-12-02636-t001:** Sequences of primers used for real-time RT-PCR.

Gene	Primer Sequence	Annealing Temperature (°C)	Product Size (bp)
Bax	5′-GCTCTGAGCAGATCAAG-3′ 5′-AGCCGCTCTCGAAGGAAGTC-3′	56	201
Bcl2	5′-TTGGAGAGCCAGTGAACAGT-3′ 5′-TGCTGATAACTGTCTGCGCT-3′	54	203
HSF1	5′-GAGCGAGGACATAAAGATTC-3′ 5′-GAGATGAGGAACTGGATGAG-3′	54	207
Glut-5	5′-TTGGAGAGCCAGTGAACAGT-3′ 5′-TGCTGATAACTGTCTGCGCT-3′	60	292
Interferon-tau	5′-ATGGCCTTCGTGCTCTCTCT-3′ 5′-AGGTCCTCCAGCTGCTGTTG-3′	55	356
Caspase-3	5′-CGATCTGGTACAGACGTG-3′ 5′-GCCATGTCATCCTCA-3′	50	359
Hsp70	5′-GACAAGTGCCAGGAGGTGATTT-3′ 5′-CAGTCTGCTGATGATGGGGTTA-3′	51	117
Dnmt3a	5′-TGATCTCTCCATCGTCAACCCT-3′ 5′-GAAGAAGGGGCGGTCATCTC-3′	54	221
β-actin	5′-GTCATCACCATCGGCAATGA-3′ 5′-GGATGTCGACGTCACACTTC-3′	56	111

**Table 2 animals-12-02636-t002:** Survival of bovine MII oocytes after vitrified-thawed (*n* = 5).

Treatment Group *	No. (%) of Oocytes	
	Thawed	Survived
0 HPC	183	138 (75.4)
10 HPC	204	164 (80.4)
50 HPC	202	170 (84.2) ^a^
100 HPC	204	154 (75.5)

* 0 HPC, VT MII oocytes treated with 0 µg/mL HPC; 10 HPC, VT MII oocytes treated with 10 µg/mL HPC; 50 HPC, VT MII oocytes treated with 50 µg/mL HPC; 100 HPC, VT MII oocytes treated with 100 µg/mL HPC. ^a^ *p* < 0.05 compared with the 0 HPC group.

**Table 3 animals-12-02636-t003:** Comparison of in vitro development between non-VT and VT groups following PA when HPC was supplemented during freezing (*n* = 5).

Treatment Group *	No. (%) of Oocytes					Total No. of Cells per Blastocyst	No. (%) of Apoptotic Cells per Blastocyst
	Thawed	Survived	PA	Cleaved (Day 2)	Blastocysts (Day 8)		
Non-VT	-	-	147	139 (94.6)	31 (21.1)	120.2 ± 6.4	2.0 ± 0,5 (1.7)
0 HPC	242	179 (74.0)	170	127 (74.7) ^a^	12 (7.1) ^a^	82.7 ± 2.5 ^a^	4.0 ± 1 (4.8)
50 HPC	201	161 (80.1)	161	121 (75.2) ^a^	21 (13.0)	100.2 ± 10.0	4.0 ± 0.7 (4.0)

* Non-VT, non-vitrified MII oocytes; 0 HPC, VT MII oocytes treated with 0 µg/mL HPC; 50 HPC, VT MII oocytes treated with 50 µg/mL HPC. ^a^ *p* < 0.05 compared with the non-VT group.

## Data Availability

The data that support the findings of this study are available upon reasonable request from the corresponding author (S.-P.P.).

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
