# Peer review of "Effect of Supplementation of Cryoprotectant Solution with Hydroxypropyl Cellulose for Vitrification of Bovine Oocytes"

_animals, 2022, doi:10.3390/ani12192636_

Round 1

Reviewer 1 Report

1.    In 2.4. Measurement of intracellular ROS In lines: 195 and 196 is: Experiments were repeated three times, with 10–20 oocytes per experiment. It is ok, but in 2.1. Oocyte preparation and in vitro maturation (IVM),  2.2. Vitrification and thawing,  2.3. PA and in vitro culture, 2.5. TUNEL assay - I don't understand how many oocytes and embryos the statistical calculations have been made of. Or, for example, in lines 251 and 304 only from the number of 5 oocytes ?.

2.    2.5. TUNEL assay

The apoptotic index was determined as the percentage of yellow blastomeres among the total number of red blastomeres. - I have a similar doubt about entering the source numerical data into the statistics?

3.      There is Figure 2 in line 260, I propose to add letters A and B to Figure 2.

Author Response

  1. Oocytes numbers used for maturation were written in line 155–156, for additional writing on other points were prepared in line 192–194 and line 215–216. The number of oocytes used in the PA embryo development experiment for each group is shown in TABLE 2.

  1. Standard deviation of the number of apoptotic cells was added to Table 3.

  1. Letters A and B have already been shown in Figure 2.

Reviewer 2 Report

In the manuscript “Effect of Supplementation of Cryoprotectant Solution with Hydroxypropyl Cellulose for Vitrification of Bovine Oocytes” the authors have used the HPC, in various doses, as cryoprotectant for bovine oocyte vitrification and evaluated its efficacy. To this aim, on bovine oocytes, subjected to vitrification or not, the production of ROS, the survival rate and the expression of some genes involved in apoptosis, stress and demethylation were evaluated; on the oocytes subjected to parthenogenesis, the developmental rate, the TUNEL assay, and the evaluation of the expression of developmental potential-related genes were determined.

This is a very interesting article, featuring well-organized experimental design. The authors' conclusions are compatible with the reported data and discussion of the work. In my opinion, there are some flaws in the Materials and Methods paragraph which, therefore, needs to be revised.

Therefore, I would suggest adding a sub-paragraph dedicated to experimental design, which describes, in addition to the methods of taking the biological material, the methods of transport and treatment (in particular the transport temperature), the number of oocytes recovered, the reasons for the choice of HPC dosages (if there are references it would be good to add them) the subdivision into experimental groups with indication, if possible, of the number of oocytes per group, and what was evaluated on the oocytes of the first part of the experiment. For the second part of the study, the one in which parthenogenesis was induced (paragraph 2.3.), I would recommend, in addition to the already present description of the induction to parthenogenesis, which determinations were made and in which experimental groups.

From L. 111 to L. 122: it would be more appropriate to use this part in the paragraph on materials and methods, I believe that in the introduction it is redundant.

L.185: "previousl" should be "previously"

Figure 2: The significance legend is absent.

Table 2: Comparison is missing after significance.

Author Response

  1. 2.1 Paragraphs of Experimental design (line 122–140) were added and explained.

  1. Paragraph 111–112 was deleted from the introduction and moved to the material and method 2–1 experimental design.

  1. L. 185 "previousl" was revised to "previously"

  1. Figure 2. the significance legend was written

  1. Table 2. was changed to Table 3 and signature displayed.

Reviewer 3 Report

In the simple summary section, you said that “the cleavage rate was significantly higher in the non-VT group”, and then you conclude that “Supplementation of vitrification solution with HPC improves the survival of VT bovine oocytes and the development capacity of embryos derived from these oocytes via PA.”. The reviewer thinks that the conclusion is not suitable.

Line 21, is VT a vitrification solution? Please show the full name if it first appears in the text. Please also check other abbreviations throughout the whole manuscript.

 In the introduction and discussion sections, you mentioned IVF and SCNT information, however; PA embryo model was performed in the current study. If possible, please explain why you used PA embryo model in your study? You can supplement this in the introduction section.

 There are at least three aims of this study showed in the introduction section. Please check and organize it again carefully. In addition, the last paragraph needs to be deleted or adds it to the discussion section appropriately.

 In the Materials and Methods section, you lost HPC detailed information. For example, how to dissolve the HPC, what is the solvent? Where do you obtain HPC? Company and product code and so on.

 Statistical Analysis section, were the percentage data tested to ensure that they are normally distributed before ANOVA?

 Table 3 (Sequences of primers used for real-time RT-PCR) appeared first in the manuscript, so renamed it as Table 1. Please check others and modified them carefully again.

 In the results section, lines 269-271, you said that “the mRNA expression level of Bcl2 was significantly lower in the 0 HPC group than in the non-VT and 10, 50, and 100 HPC groups.” However, there was an incorrect “*” shown in Figure 3B. 

Author Response

  1. Summary part conclusion (Line 28-31) was revised.

  1. The abbreviation for VT on line 21 was explained.

  1. Points to be corrected were revised in lines 71 to 82 of the introduction paragraphs and the contents were added.

Round 2

Reviewer 2 Report

The authors added, as suggested, almost all the required information. There are still some clarifications to be made, probably because I have not been clear and which I report below.

Paragraph 2.4.: I previously suggested to list the measurements you did (mRNA expression levels of some genes?) in the groups L.131: you write “about 200 oocytes”, but it would be more correct to write respectively the real number of oocytes for each experimental group and it is important not to be approximate, these same numbers have been used for statistical aims, so you can get them easily from tables.

L. 134: see previously comment.

l. 231: again "previousl" should be "previously”.

Figure 2: between which groups was significance measured? *p<0.05 versus …?

Figure 5: p value of ** is absent.

Table 2: “a p < 0.05” compared to Thawed?

Author Response

Thank you for the review.

1. The number of oocytes and blastocysts used in the experiment was detailed in 2.7 (lines 237 to 238)

2. The word "previousl" was rewritten as "previously". (line 235)

3. Figure 2 was revised, and I wrote about the meaning of *.

4. Figure 5: p value of ** was described. 

5. Table 2: p value of a was described. 
